

# Conspecific and heterospecific cueing in shelter choices of *Blaptica dubia* cockroaches

Todd M. Freeberg[1,2], S. Ryan Risner[1], Sarah Y. Lang[1] and Sylvain Fiset[3]

[1] Department of Psychology, University of Tennessee—Knoxville, Knoxville, Tennessee, United States
[2] Department of Ecology & Evolutionary Biology, University of Tennessee—Knoxville, Knoxville, Tennessee, United States
[3] Secteur Sciences Humaines, Université de Moncton—Edmundston, Edmundston, New Brunswick, Canada

Corresponding author
Todd M. Freeberg, tfreeber@utk.edu

## ABSTRACT

**Background:** Like many cockroaches, Argentinian wood roaches, *Blaptica dubia*, prefer darker shelters over lighter shelters. In three experiments, we asked whether chemical cues from other roaches might influence shelter choice, a process known as conspecific or heterospecific cueing, depending on whether the cues come from an individual of the same or a different species, respectively.

**Methods:** Each experiment involved trials with focal *B. dubia* cockroaches in testing arenas containing plastic shelters of varying levels of darkness, with filter paper under each shelter acting as a carrier for chemical cues. In Experiment 1, we tested female and male *B. dubia* cockroaches with two shelters matched for darkness but differing in cues (conspecific *vs.* none). The shelter with no cue contained a blank filter paper as a control. In Experiment 2 (conspecific cueing) and Experiment 3 (heterospecific cueing), we tested *B. dubia* cockroach choices for lighter or darker shelters with filter papers containing chemical cues of other roaches or no chemical cues. For the conspecific cueing study of Experiment 2, we used chemical cues from other *B. dubia* cockroaches. In contrast, for the heterospecific cueing study of Experiment 3, we used chemical cues from a different species, the death's head cockroach, *Blaberus craniifer*.

**Results:** In Experiment 1, *B. dubia* cockroaches overwhelmingly preferred shelters with conspecific chemical cues over darkness-matched shelters without cues. In Experiments 2 and 3, they strongly preferred darker shelters, especially when chemical cues were present. Additionally, they were more likely to be under the lighter shelter when chemical cues were present there. These results reveal that the public information *B. dubia* cockroaches gain from chemical cues—including those from other species—can drive shelter choices in this species.

## INTRODUCTION

Living in groups provides numerous benefits to social species, such as a greater ability to avoid predation and to utilize resources important to survival (*Krause & Ruxton, 2002*).

This has been demonstrated in hissing cockroaches, *Gromphadorhina portentosa*, with group-living individuals conserving water more effectively over long periods of time than their isolated counterparts (*Yoder & Grojean, 1997*). Similar survival benefits of group living have been identified in sand fiddler crabs (*Uca pugilator*; *Yoder et al., 2005*). In addition to benefits related to resource use, social behavior is well known to improve individuals' ability to obtain food. For example, when compared to single individuals, groups of the caterpillar *Mechanitis menapis* more effectively overcome the chemical defenses of the nightshade *Solanum acerifolium* to eat its leaves (*Despland, 2019*).

Individuals often make decisions about resources in contexts involving social cues. Habitat decisions of many songbird species, for example, are affected by the songs and calls of conspecifics coming from certain areas within the habitat (*e.g.*, American redstarts, *Setophaga ruticilla*: *Hahn & Silverman, 2006*; grasshopper sparrows, *Ammodramus savannarum*: *Andrews, Brawn & Ward, 2015*). Similarly, male desert grasshoppers, *Ligurotettix couilletti*, are attracted to bushes containing either live male conspecifics or playbacks of male conspecific calls (*Muller, 1998*). Male strawberry poison frogs, *Oophaga pumilio*, select and establish territories in areas with conspecific territorial males, likely based on their aposematic visual signals (*Folt, Donnelly & Guyer, 2018*). Black-and-white ruffed lemur mothers, *Varecia variegate*, base their nest site decisions on the presence of other mothers' nest sites (*Baden, 2019*). German cockroaches, *Blatella germanica*, tested on a series of Y-shaped stems, chose stems containing chemical cues of their own group compared to unmarked stems (*Jeanson & Deneubourg, 2006*). This process of basing activity in part on the signals and cues of conspecifics is referred to as conspecific cueing (*Stamps, 1988*). Such cueing has additionally been shown to serve an important role in local enhancement about foraging patches (*e.g.*, honeybees, *Apis mellifera*: *Lowell et al., 2019*). However, the social behavior of some species sometimes depends on heterospecific cues. For example, *Nasutitermes corniger* termites prefer food sites with chemical cues of conspecifics of other colonies, or even heterospecific termites, over food sites containing their own colony's chemical cues (*Silva et al., 2021*).

Other factors have also been shown to influence habitat choices in animal species. Many species of cockroach prefer darker habitats over lighter habitats when considering potential shelter sites (*Turner, 1912*; *Bell, Roth & Nalepa, 2007*). Groups of adult American cockroaches, *Periplaneta americana*, strongly prefer darker shelters over structurally matched but lighter shelters (*Canonge, Deneubourg & Sempo, 2011*). Beyond the darkness of shelters, cockroaches likely attend to conspecific cues or signals in those shelters (*Bell, Roth & Nalepa, 2007*). For example, robotic cockroaches coded to prefer a lighter shelter over a darker shelter increased the likelihood of American cockroaches choosing the lighter over the darker shelter due to those artificial tactile cues (*Halloy et al., 2007*). German cockroaches, *Blatella germanica*, are more likely to aggregate at sites containing cuticular hydrocarbons of members of their own species (*Rivault, Cloarec & Sreng, 1998*) and at sites containing chemical cues of members of their own group compared to those of unfamiliar conspecific groups (*Rivault & Cloarec, 1998*). These chemicals—and preferences for them and their concentrations regarding shelter choice—vary across species of *Periplaneta* (*Saïd et al., 2005*). Additionally, German cockroach nymphs prefer

to aggregate near an adult female (even when dead and cleaned of cuticular hydrocarbons) compared to a comparable site with filter paper containing conspecific cuticular hydrocarbons (*Hamilton, Wada-Katsumata & Schal, 2019*). Individuals of this species are also more likely to remain at a site if one or more conspecifics are already present at that site (*Ame, Rivault & Deneubourg, 2004*). Clearly, individual cockroaches are sensitive to signals and cues of conspecifics in a range of modalities.

We recently found that, like the case for *P. americana* described above (*Canonge, Deneubourg & Sempo, 2011*), groups of Argentinian wood roaches, *Blaptica dubia*—hereafter, *B. dubia* cockroaches, preferred darker shelters over lighter shelters and the effect was stronger for larger groups of 10 or 15 individuals in comparison to groups of five individuals (*Freeberg & Fiset, 2023*). Here, we tested for effects of conspecific and heterospecific chemical cues on shelter choice in *B. dubia* cockroaches in three separate experiments. We do not yet know what the specific chemical cues are, though given the method we used they are likely cuticular hydrocarbons, chemicals in feces, or both. In all three experiments, we also assessed whether there were sex differences in chemical cue-guided shelter choices. Female *B. dubia* cockroaches are heavier than males, and sex differences in movement and in learning spatial layouts of habitat exist in some cockroach species (*e.g.*, *Periplaneta orientalis*: *Turner, 1912*; *P. fuliginosa*: *Appel & Rust, 1986*).

In Experiment 1, we tested individual shelter choices in arenas containing two shelters with identical features and illuminance levels, but that differed in the chemical cues available on filter papers under the shelters. The experimental choice was between filter paper with or without conspecific cues. We expected the *B. dubia* cockroaches to choose conspecific cues over no cues, as this effect has been documented in a wide range of species and modalities (*e.g.*, acoustic signals in songbirds: *Farrell et al., 2012*; chemical cues in cockroaches: *Rivault, Cloarec & Sreng, 1998*, fish: *Golub, Vermette & Brown, 2005*, and rodents: *Davis, 1970*; presence of conspecific offspring in frogs: *Rudolf & Rodel, 2005*).

In Experiment 2, we provided a lighter and a darker shelter in each testing arena. For half of the *B. dubia* cockroaches in the trials, the lighter shelter contained a filter paper with conspecific cues and the darker shelter contained a filter paper without conspecific cues. This was reversed for the other half of the individuals in the trials. We knew from our earlier study (*Freeberg & Fiset, 2023*) that groups of *B. dubia* cockroaches preferred darker shelters over lighter shelters (as has been demonstrated in a wide range of cockroach and wood roach species; *e.g.*, *Bell, Roth & Nalepa, 2007*) and here we sought to test whether this preference-for-shelter could be influenced by chemical cues from conspecifics. If shelter choice in this species is primarily driven by variation in shelter darkness, we predicted no effect of conspecific cues and a main effect of shelter darkness. If, however, shelter choice is driven in part by conspecific cues, we predicted preferences for both the darker and the lighter shelters to be stronger if those shelters contained filter papers with conspecific cues.

In Experiment 3, we largely replicated the methodology of Experiment 2, using filter papers containing either chemical cues from heterospecifics or no chemical cues. The heterospecific chemical cues were obtained from a breeding colony of death's head cockroaches, *Blaberus craniifer*. In nature, these two species would not come into contact with one another, as *B. dubia* occurs in South America and *B. craniifer* occurs in Mexico

and Central America. Thus, chemical cues of *B. craniifer* in Experiment 3 represent species-level novel cues for *B. dubia* individuals. Nonetheless, it has recently become clear that mixed-species aggregations of arthropods are fairly common (though understudied compared to similar mammalian and avian groups; *Boulay et al., 2019*; *Goodale, Beauchamp & Ruxton, 2017*; *Schaller et al., 2018*) and so research is needed on the factors influencing the formation and maintenance of those aggregations.

## MATERIALS AND METHODS

### General methods

The adult *B. dubia* cockroaches tested came from laboratory breeding populations that were originally obtained from different mail order companies (see *Freeberg & Fiset, 2023*). The female and male subjects tested were randomly drawn from these breeding populations. Breeding populations were maintained in large plastic containers (50 cm × 36 cm × 30 cm) with a screen mesh top, under a 12 h light/12 h dark cycle. Each large plastic container included several egg cartons (29 cm × 29 cm), with an *ad libitum* mix of dry cat and dog food as well as water crystals. Containers were stocked with fresh fruit and vegetable pieces (typically melon, papaya, apple, or carrot) once or twice per week. Breeding containers were housed within Med Associates, Inc. (St. Albans, VT, USA) Large Monkey cubicles and were typically maintained between 26 and 31 °C and between 50% and 70% humidity. Light levels in the test arenas were assessed with a Sekonic i-346 illuminometer (Tokyo, Japan), with multiple measurements taken in the open parts of the test arenas and under the two shelters (see Table 1). Once *B. dubia* cockroaches were placed into the test arenas, they were not disturbed by the researchers for the duration of the trials until after their shelter choice was coded (see below).

Individuals tested for shelter choice were obtained from the above-mentioned breeding containers. After use in shelter choice tests, these individuals were placed into another large container (the "retirement" population) so that they were not tested again. To obtain chemical cues in these experiments, filter papers (11 cm diameter, #415, VWR International) were placed inside either the retirement population container containing *B. dubia* cockroaches (Experiments 1 and 2) or inside a separate container holding a breeding population of death's head roaches, *B. craniifer* (Experiment 3).

### Shelter choice codings and inter-observer reliability

In all three experiments, the individual shelter choices were coded as: under one of the shelters, under the other shelter, or under no shelter (out in the test arena). We tested inter-observer reliability (*Burghardt et al., 2012*) for 35 randomly-selected groups across the three experiments by having a main observer and a trained research assistant concurrently and independently code the location of each individual as under shelter or out in the test arenas. Inter-observer reliability was high (Cohen's kappa statistics and agreement for the cued shelter (first experiment) or the darker shelter (second and third experiment) $\kappa = 0.96$, 34/35 agreements; non-cued shelter or lighter shelter $\kappa = 0.93$, 34/35 agreements; no shelter $\kappa = 0.88$, 33/35 agreements).

**Table 1 Shelter lighting differences and chemical cues information in the three experiments.** Light measurements (lux) under and outside of shelters and number of days filter papers were in *B. dubia* (Experiment 1 and 2) or *B. craniifer* (Experiment 3) colonies.

| Experiment | Ranges in light levels (lux) under/outside of shelters | | | Filter paper days | |
|---|---|---|---|---|---|
| | Darker shelter | Lighter shelter | Outside | Median | Range |
| 1 | 8.2–10.1* | | 449–570 | 5 | 3–14 |
| 2 | 103–139 | 262–418 | 559–599 | 6 | 3–13 |
| 3 | 9.3–13.3 | 101–127 | 531–618 | 7 | 3–14 |

**Note:**
  * Two darkness-matched shelters used.

### General statistical approach

Given that our outcome variable—which was the location of the *B. dubia* cockroach at the end of the trial—was a nominal variable with three unordered choice categories (under one of the shelters, under the other shelter, or under no shelter (out)), we modelled it *via* a multinomial distribution (*Hardin & Hilbe, 2018*; *Kaufman, 2019*). In a multinomial model, one response category is designated as the reference category, and the others are compared to it (*Long & Freese, 2015*). The shelter with the conspecific cue was chosen as the reference category in Experiment 1, while the darker shelter was selected as the reference category in Experiments 2 and 3.

We used the statistical software Stata 16.1 (*StataCorp, 2019*) to analyze our data. Because our data had a hierarchical structure, we analyzed our data using a series of multilevel multinomial models. With Stata 16.1, multilevel multinomial models are performed *via* a generalized structural equation approach, which uses a multinomial distribution with a nonlinear logit link function. As *Long & Freese (2015)* described, multinomial logistic models fit separate multilevel binary logistic regression models for each category of the outcome variable relative to the reference category. In the current study, our multinomial model comprised one sub-equation for each of the two non-reference *B. dubia* cockroach choices, that is, one for the *control cue shelter* (Experiment 1) or the *lighter shelter* (Experiments 2 and 3) and one for *out*. Furthermore, we estimated the likelihood of the outcome variable using the adaptive Gaussian quadrature method, which is known for being highly accurate (*Garson, 2020*).

As suggested by *Kaufman (2019)* for nonlinear models, especially when interactions are involved, we compared our different statistical models using a series of likelihood ratio (LR) tests. An LR test compares a full model with a nested model that includes a subset of the variables in the full model. The null hypothesis of the LR test is that the nested model fits the data as well as the full model. Thus, when the nested model is tested against the full model, if the *p*-value of the LR test is equal to or less than 0.05 (we selected a *p*-value of 0.05 to reject the null hypothesis), it means that the full model fits the data better than the nested model.

For each experiment, a series of LR tests were conducted, comparing a reduced model (the nested model) to a more complex model (the full model). The first tests were

performed using null models that did not have any predictors, to evaluate whether the nested structure of the data should be considered. Once the appropriate random structure was established, we utilized a backward stepwise comparison methodology, beginning with the full model consisting of all predictors and interactions, and subsequently removing a single term from the regression equation to form the reduced model. If the full model did not fit the data better than the reduced model, the subsequent model comparison treated the reduced model as the full model, and we removed an additional term from the regression equation to create the next reduced model. This stepwise approach was repeated until the full model fit the data better than the reduced model, or until the null model fit the data better than the fixed model.

After selecting the most parsimonious model, in Experiment 1, we used Stata *lincom* command (for a justification, see *Mize, Doan & Long, 2019*) to compare the probabilities of finding a *B. dubia* cockroach under the shelter with the conspecific cue to the probability of finding a *B. dubia* cockroach under the shelter with the control cue. In Experiments 2 and 3, to assess how conspecific cue location affected the probability of detecting individuals in the darker shelter, lighter shelter, or out, we computed average marginal effects (AMEs). AMEs compute the average discrete change in probability from the base level to the next higher level of each categorical predictor (*Long & Freese, 2015*; *Williams, 2012*). In Experiments 2 and 3, the darker shelter served as the reference level for the conspecific cue location, and AMEs were used to compare individuals' responses to the conspecific cue filter paper when it was placed under the lighter shelter.

## First experiment: effect of conspecific cues under darkness-matched shelters

Individuals were tested from February to May 2019 in 24 groups. Each group comprised two adult females and two adult males. Individuals in each group were given a novel color of acrylic paint near the wing base of both wings for means of individual identification. Each group was housed in a smaller plastic container (35 cm × 21 cm × 12 cm; hereafter, "group housing") with small holes drilled for ventilation, smaller stacked egg carton pieces (~15 cm × ~11 cm), *ad libitum* dry cat and dog food, and water crystals. Fresh fruit or vegetable pieces were added once or twice per week. Individuals housed in the group housing containers were kept in a Med Associates, Inc. large monkey cubicle separated from the breeding colony containers.

Groups were tested separately in four large plastic containers (Sterlite Clear View 104 L latch containers; 88 cm × 48 cm; hereafter, "test arenas") each containing two shelters matched for size and light intensity. Shelters were 20 cm × 25 cm clear acrylic sheets held 4 cm off the floor of the test arenas by plastic spacers at the corners. Each shelter was covered with a 15 cm × 23 cm black foam sheet positioned symmetrically within the dimensions of the acrylic sheet. Shelters were positioned at either end of the test arenas at 5 cm from the short sides and 7 cm from the long sides. The spaces under the shelters were considerably darker than the spaces outside of shelter in the test arenas (Table 1).

To provide chemical cues for each trial, a filter paper was secured with tape to the test arena floor underneath each shelter, roughly 30 min before test subjects were placed in the

test arena. One filter paper had been kept in the large retirement colony container for at least 3 days (Table 1). This filter paper was the "conspecific cue" stimulus, and typically had soiling from, and likely contained cuticular hydrocarbons of, *B. dubia* cockroaches on it when used during testing. The second filter paper was taken directly from its packaging box and represented the control stimulus. Conspecific cue and control stimulus filter papers were counter-balanced across shelters, test groups, and test arenas over the course of the study.

The four test arenas were placed on top of a long table so that four groups of *B. dubia* cockroaches could be tested at once (one in each test arena). On either end of the test arena table stood lamps that were linked to a ramp-up timer. This ensured that illumination began at 11:30 am (EST) and gradually increased in intensity over 30 min to full light intensity. We did this gradual ramping up of light intensity to mimic natural changes in lighting and to help ensure individuals had sufficient time in darker conditions to explore the test arenas. Individuals were placed into the center of testing arenas by 11:15 am each day of testing. At ~12:05 pm on testing days, we turned on the ceiling lights in the testing room (to further increase light intensity) and waited 3 min before coding the shelter choices of individuals of each group in the four test arenas. Our three codings were: under the shelter with conspecific cue filter paper, under the shelter with control paper, or under no shelter (out in the test arena). After coding of shelter choices, individuals were returned to their group housing containers and cubicle and the testing arenas were cleaned with odorless baby wipes (Comforts fragrance-free baby wipes; Kroger, Cincinnati, OH, USA; this method was validated to remove chemical cues of roaches in *Freeberg & Fiset, 2023*). We tested shelter choices of each group of *B. dubia* cockroaches six times, with at least 3 days for each group between consecutive tests.

### First experiment: statistical analyses

Data of Experiment 1 had a nested structure. For each of the six trials, the *B. dubia* cockroaches were tested in groups of four (two males and two females) for a total of 24 groups. Thus, to assess the variability associated with the three levels of this nested structure (trials > individuals > groups), we ran a series of multilevel multinomial logistic regression models in which we treated individuals and their respective group as random factors. Then, we ran additional multinomial models to investigate if the fixed effects, sex and trial (and the interaction), could predict the individual's shelter choice relative to the location of the conspecific cue.

## Second experiment: effect of conspecific cues under dark *vs.* light shelters

From November 2020 to May 2021 we tested sixteen rounds of new *B. dubia* cockroaches. Within each round we carried out four trials to counter-balance locations of darker and lighter shelters as well as locations of chemical cues under the shelters. In line with this setup, the first round of testing involved the following order: (1) darker shelter, left side of test arenas, with chemical cues; (2) lighter shelter, left side of test arenas, with chemical cues; (3) darker shelter, right side of test arenas, with chemical cues; and (4) lighter shelter,

right side of test arenas, with chemical cues. The chemical cues were provided *via* filter paper using the same methodology as described in Experiment 1. The filter papers were left in the *B. dubia* retirement colony for at least 3 days before use in the experiment as conspecific cues (Table 1).

For each trial within a round, four females and four males were drawn from the breeding colony. For each sex, we tested a solitary individual ($N = 1$) and a small group ($N = 3$) separately. Each set was tested in the large plastic container test arenas as described above for the first experiment with three methodological differences. The first difference was that one of the shelters had dark blue or purple plastic sheeting on top of it (darker shelter) and the other had light yellow or orange plastic sheeting on top of it (lighter shelter; Avery insertable plastic dividers, Brea, California). Both the darker and lighter shelters were considerably darker than the spaces outside of shelters in the test arenas, and the darker shelters were darker than the lighter shelters (Table 1). The second difference was that we conducted the entire study in a bright room—from the time of set up of the *B. dubia* cockroaches to the coding of their shelter choices. We chose not to ramp up the light levels in the room as the method was more practically limiting and did not seem to result in different patterns of shelter choice in comparison to different bright light regimes (as in *Freeberg & Fiset, 2023*). The third difference was that we coded individuals' shelter choices 2 h after they were placed into the test arenas. After testing was completed, each individual was placed into the retirement colony container.

### Second experiment: statistical analyses

We used the same statistical approach as the one we developed in the first experiment. Although the individuals were not tested multiple times as in the first experiment, our data had a nested structure: females and males were separately nested into a different round of testing for a total of sixteen clusters. To better capture the variability associated with this two-level structure (individuals > rounds), we ran a series of multinomial logistic regression models. Firstly, we examined the impact of the nested structure of our data and secondly, we investigated if the fixed effects (the conspecific cue location, group size, sex, and their interactions) could predict *B. dubia* cockroaches' shelter choice. The number of days the filter paper was kept in the colony did not significantly affect the results, so we excluded this covariable from our statistical models to simplify the analysis.

### Third experiment: effect of heterospecific cues under dark *vs.* light shelters

From February to June 2022 we tested sixteen rounds of new *B. dubia* cockroaches. This study was conducted largely using the methodology of the second experiment, with the key difference being that the chemical cues were heterospecific rather than conspecific, obtained from a breeding colony of death's head roaches. Our colony of death's head roaches developed from a starter group obtained from Cape Cod Roaches (capecodroaches.com). Each set of *B. dubia* cockroaches was tested in large plastic container test arenas as described above for the second experiment, except that the lighter shelters in the third experiment were the dark blue or purple plastic sheets and the darker

shelters in the third experiment had opaque black plastic sheets. Darker and lighter shelters were considerably darker than the spaces outside of shelters in the test arenas, and the darker shelters were darker than the lighter shelters (Table 1). Filter papers for chemical cues in the study were left in the *B. craniifer* breeding colony for at least 3 days before use in the experiment (Table 1).

### Third experiment: statistical analyses

Because this experiment was methodologically similar to that of the second experiment, except for the nature of the cue placed under the shelters (heterospecific rather than conspecific cueing) and the different light regimes under the shelters, we adopted the same approach as the one used in the previous experiment. As in Experiment 2, the number of days the filter paper was kept in the colony did not significantly affect the results and we excluded this covariable from our statistical models.

### Statistical analyses for all three experiments: access to data

Our data sets, Stata scripts, and analyses can be found on the Open Science Framework at https://osf.io/3y79n/?view_only=3ae061c93b164f3c8737d9d636b3fb46.

## RESULTS

### First experiment: effect of conspecific cues under darkness-matched shelters

Three *B. dubia* cockroaches from different groups died during the study. One male died after three trials, and one male and one female died after their first trial. All three were replaced by new individuals to complete the six trials. To simplify the analyses, we removed the data of the three individuals that died during the experiment and those of the three substitutes. Thus, we analyzed the shelter choice of 93 individuals ((2 sexes × 48 individuals)—3 deceased individuals) for 558 observations (each included individual was tested six times under the same condition: see Table S1). Altogether, the *B. dubia* cockroaches selected the conspecific cue shelter 60.57% of the time and the control cue shelter about 10.93%. We found 28.49% of the individuals under no shelter (out). However, because each individual was tested six times, these percentages could change from trial to trial for each individual of both sexes. Our multilevel multinomial statistical models allowed us to examine these possible variations (see below).

### Model selection

Model 2, which included individuals as random effects (see Table 2), was the most efficient null model. An LR test revealed that Model 2 offered better data fit compared to Model 1 ($X_2(2) = 6.24$, $p = 0.044$) and adding groups as random effects (Model 3) did not improve the fitting of the data ($X_2(2) = 0.00$, $p = 0.999$). As a result, we included individuals as random effects into our fixed effects models. In terms of fixed effects, no model proved more suitable than Model 2 (LR tests, $p > 0.05$). Thus, with regard to sex and trial number, the introduction of such factors did not enhance the model's likelihood. Ultimately, Model 2 (a null model) was deemed the most parsimonious model. The conclusion is well-illustrated by Fig. 1 (using the predictions of Model 6), indicating that the probability

**Table 2 Model comparison using log-likelihood, AIC and BIC in Experiment 1.**

| Model | Type | Random intercept effects | Fixed effects | Log likelihood | Number of parameters | Degrees of freedom | AIC | BIC |
|---|---|---|---|---|---|---|---|---|
| 1 | Null | — | — | −504.084 | 3 | 2 | 1,012.167 | 1,020.816 |
| 2* | Null | Roaches | — | −500.964 | 7 | 4 | 1,009.929 | 1,027.226 |
| 3 | Null | Roaches > Groups | — | −500.964 | 11 | 5 | 1,011.929 | 1,033.550 |
| 4 | Full | Roaches | Sex | −499.941 | 11 | 6 | 1,011.882 | 1,037.829 |
| 5 | Full | Roaches | Sex + Trial | −499.450 | 13 | 8 | 1,014.900 | 1,049.495 |
| 6 | Full | Roaches | Sex × Trial | −497.805 | 17 | 10 | 1,015.610 | 1,058.854 |

Notes:
> Nested in.
* Selected model

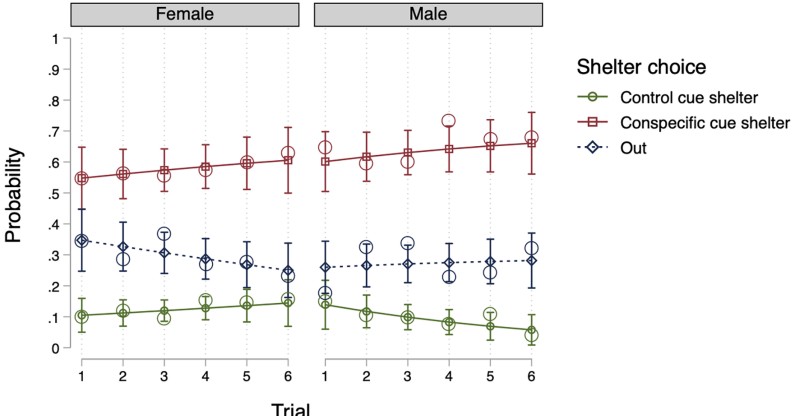

**Figure 1 Predicted probability for shelter choice by sex and trial in Experiment 1.** The open circles represent the observed mean proportions, the horizontal lines represent the predicted means, and the error bars represent the 95% confidence intervals of the predicted mean.

of selecting a shelter was comparable for both males and females, and the same across all trials.

### Selected model

Table S2 (see Supplemental Materials) shows the estimated intercepts for each shelter choice of Model 2 in Experiment 1, which we expressed as relative risk ratios – a direct measure of effect size for nonlinear regression models (*Ferguson, 2009*). The risk of finding a *B. dubia* cockroach under the control cue shelter ($b_0 = 0.177$, $Z = −9.42$, $p < 0.001$, 95% CI [0.124–0.254]) or out ($b_0 = 0.433$, $Z = −6.17$, $p < 0.001$, 95% CI [0.332–0.565]) was significantly lower relative to the risk of finding an individual under the shelter with the conspecific cue. These estimates support the conclusion that *B. dubia* cockroaches were primarily found under the shelter with the conspecific cue.

As suggested by *Long & Freese (2015)* for multinomial models, we also examined the probabilities predicted by our selected model. Overall, Model 2 estimates that the probability of finding an individual under the shelter with the conspecific cue was 60.57% (95% CI [55.66–65.48]), under the shelter with the control cue was 10.94% (95% CI [8.33–13.55]), and outside of a shelter in the open test arena was 28.49%

**Table 3 Observed frequencies (and percentages) for each of the three locations where the *dubia* roaches were found as a function of sex, group size and the location of the conspecific cue filter paper (cue) in Experiment 2.** For simplicity, interactions between the predictors are not included. The percentage of individuals found at the three possible locations was similar for each level of sex and group size. However, the percentages changed drastically when the cue location changed from the darker shelter to the lighter shelter (see the Results section).

| Location where individuals were found | Sex | | Group size | | Cue location | |
|---|---|---|---|---|---|---|
| | Females | Males | 1 | 3 | Darker shelter | Lighter shelter |
| Darker shelter | 188 (73.44%) | 180 (70.31%) | 93 (72.66%) | 275 (71.61%) | 241 (94.14%) | 127 (49.61%) |
| Lighter shelter | 35 (13.67%) | 51 (19.92%) | 19 (14.84%) | 67 (17.45%) | 5 (1.95%) | 81 (31.64%) |
| Out | 33 (12.89%) | 25 (9.77%) | 16 (12.50%) | 42 (10.94%) | 10 (3.91%) | 48 (18.75%) |

(95% CI [24.09–32.89]). Interestingly, these estimates corroborate the percentages of observed choices we first presented. Moreover, the probability of finding a *B. dubia* cockroach under the shelter with the conspecific cue was higher than the probability of finding an individual under the shelter with the control cue (0.496, $Z = 14.92$, $p < 0.001$, 95% CI [0.431–0.562]).

## Second experiment: effect of conspecific cues under dark *vs.* light shelters

We analyzed the shelter choice of 512 *B. dubia* cockroaches (16 rounds ∗ 32 individuals/round; see Table S3). We found 71.88% of the individuals under the darker shelter, 16.80% under the lighter shelter, and 11.33% under no shelter. However, these percentages must be nuanced as a function of the location of the conspecific cue, group size, and sex.

The percentages of individuals found under the darker shelter, lighter shelter, or out were similar, whatever the group size or sex (Table 3). However, the location of the conspecific cue filter paper impacted individuals' shelter choice strongly. The proportion of individuals found under the darker shelter was higher when the conspecific cue was located under the darker shelter as opposed to under the lighter shelter. Conversely, compared to the reference of the darker shelter, the proportion of individuals found under the lighter shelter was higher when the conspecific cue was placed under the lighter shelter as opposed to under the darker shelter. Therefore, we ran a series of multinomial logistic regression models to confirm these observations and examine the possible interaction between these factors.

### Model selection

Model 2 constituted the most parsimonious null model (see Table 4). Random effects for rounds of testing were included in this model and it exhibited superior data fitting relative to Model 1 ($X_2(2) = 6.24$, $p = 0.044$). Accordingly, rounds of testing were incorporated into our fixed effects models. Concerning fixed effects, Model 3 was the most parsimonious fixed effects model (LR Tests, $p$s > 0.05) and compared to Model 2 (a null model), it demonstrated better data fitting ($X_2(2) = 148.44$, $p < 0.001$). Thus, Model 3 emerged as the preferred model. Including the location of the filter paper with the conspecific cue as a predictor improved the predictions of our statistical model. This finding supports our initial observations (see Table 3), suggesting that the location of the conspecific cue

**Table 4 Model comparison using log-likelihood, AIC and BIC in Experiment 2.**

| Model | Type | Random intercept effects | Fixed effects | Log likelihood | Number of parameters | Degrees of freedom | AIC | BIC |
|---|---|---|---|---|---|---|---|---|
| 1 | Null | — | — | −401.268 | 3 | 2 | 806.536 | 815.013 |
| 2 | Null | Rounds | — | −400.952 | 7 | 4 | 804.144 | 821.097 |
| 3* | Full | Rounds | Cue | −329.242 | 11 | 6 | 659.708 | 685.138 |
| 4 | Full | Rounds | Cue + Group | −328.918 | 15 | 8 | 663.007 | 696.913 |
| 5 | Full | Rounds | Cue × Group | −326.628 | 23 | 10 | 662.367 | 704.750 |
| 6 | Full | Rounds | Cue × Group + Sex | −324.316 | 27 | 12 | 661.321 | 712.181 |
| 7 | Full | Rounds | Cue × Group × Sex | −320.773 | 59 | 18 | 666.217 | 742.507 |

Notes:
Cue, Conspecific cue location; Group, Group size.
\* Selected model.

significantly affected the shelter choice behavior of *B. dubia* cockroaches. Neither sex nor group size (or their interaction) had any effects on our outcome variable's predictions.

### Selected model

We report the relative risk ratios of Model 3 for cue location in Table S4. When we compare the conspecific cue filter paper under the lighter as opposed to the darker shelter, the risk of finding the individual under the lighter shelter increased by 36.02 (95% CI [15.97–81.25]) and the risk of finding the individual in any shelter increased by 9.11 (95% CI [3.26–25.45]). Although these risk ratios suggest a strong effect of the conspecific cue location on *B. dubia* cockroaches' shelter choice, *Long & Freese (2015)* pointed out that the coefficients of multinomial models, especially those associated with the predictors, are challenging to understand. Indeed, in these statistical models, there is no estimate associated with the base category, which, in our case, was the darker shelter.

To better understand the impact of the conspecific cue location on the probability of finding individuals under the darker shelter, the lighter shelter, we calculated the AMEs for cue location for each shelter choice (Fig. 2). On average, placing the conspecific cue under the lighter shelter rather than under the reference darker shelter decreased an individual's probability of being under the darker shelter by 0.445 (95% CI = [−0.539 to −0.352], $Z = -9.37$, $p < 0.001$), but increased its probability of being under the lighter shelter by 0.297 (95% CI [0.212–0.382], $Z = 6.81$, $p < 0.001$). This manipulation also increased an individual's probability of being under no shelter by 0.148 (95% CI [0.081–0.215], $Z = 4.34$, $p < 0.001$). Thus, the presence of conspecific cues under the lighter shelter strongly decreased the probability of finding *B. dubia* cockroaches under the darker shelter but significantly increased the probability of finding them under the lighter shelter or under no shelter at all.

### Third experiment: effect of heterospecific cues under dark *vs.* light shelters

As in the second experiment, we analyzed the shelter choice of 512 *B. dubia* cockroaches (see Table S3). Overall, 83.40% of the individuals were found under the darker shelter, 5.27% under the lighter shelter, and 11.33% under no shelter.

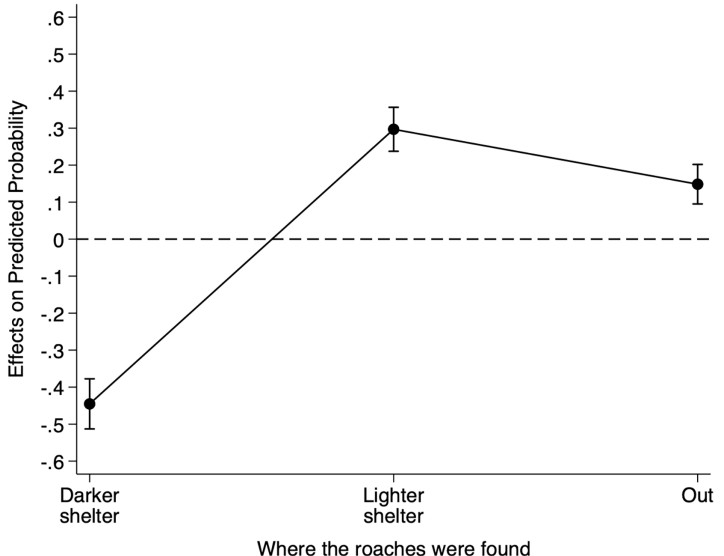

**Figure 2 Discrete change of conspecific cue location on the probability that *dubia* roaches were in the darker shelter, lighter shelter, or not in any of them (Out) in Experiment 2.** Discrete change (average marginal effects) of conspecific cue location (from the reference darker shelter to the lighter shelter) on the probability that *dubia* roaches were in the darker shelter, lighter shelter, or not in any of them (Out) in Experiment 2. The dots represent the average marginal effects, and the error bar represents the 95% confidence intervals. The dashed line represents the absence of an effect.

**Table 5 Observed frequencies (and percentages) for each of the three locations where the dubia roaches were found as a function of sex, group size and the location of the heterospecific cue filter paper (cue) in Experiment 3.** For simplicity, interactions between the predictors are not included.

| Location where individuals were found | Sex | | Group size | | Cue location | |
|---|---|---|---|---|---|---|
| | Females | Males | 1 | 3 | Darker shelter | Lighter shelter |
| Darker shelter | 209 (81.64%) | 218 (85.16%) | 99 (77.34%) | 328 (85.42%) | 228 (89.06%) | 199 (77.73%) |
| Lighter shelter | 14 (5.47%) | 13 (5.08%) | 9 (7.03%) | 18 (4.69%) | 2 (0.78%) | 25 (9.77%) |
| Out | 33 (12.89%) | 25 (9.77%) | 20 (15.62%) | 38 (9.90%) | 26 (10.16%) | 32 (12.50%) |

Table 5 presents the frequencies (and percentages) of individuals' shelter choices for each level of the heterospecific cue location, group size, and sex. Overall, shelter choice does not seem to be affected by the sex of the individual. The proportion of individuals found under the darker shelter, lighter shelter, or out was almost identical for females and males. Regarding group size, the proportion of individuals found under the darker shelter appears slightly higher when there were three individuals in a group than when an individual was alone. This observation, however, remains to be confirmed by our statistical analyses. Finally, the location of the heterospecific cue filter paper seems to impact individuals' shelter choice more strongly. The proportion of individuals found under the darker shelter was higher when the heterospecific cue was located under the darker shelter than under the lighter shelter. We ran a series of multinomial logistic regression models to

**Table 6 Model comparison using log-likelihood, AIC and BIC in Experiment 3.**

| Model | Type | Random intercept effects | Fixed effects | Log likelihood | Number of parameters | Degrees of freedom | AIC | BIC |
|-------|------|--------------------------|---------------|----------------|----------------------|--------------------|------|-----|
| 1 | Null | — | — | −283.282 | 3 | 2 | 570.564 | 579.041 |
| 2 | Null | Round | — | −281.708 | 7 | 4 | 571.416 | 588.369 |
| 3* | Full | — | Cue | −270.400 | 7 | 4 | 548.800 | 565.754 |
| 4 | Full | — | Cue + Group | −268.204 | 11 | 6 | 548.409 | 573.839 |
| 5 | Full | — | Cue × Group | −267.283 | 19 | 8 | 550.566 | 584.473 |
| 6 | Full | — | Cue × Group + Sex | −266.605 | 23 | 10 | 553.210 | 595.593 |
| 7 | Full | — | Cue × Group × Sex | −264.537 | 55 | 16 | 561.074 | 628.887 |

Notes:

Cue, Heterospecific cue location; Group, Group size.

* Selected model.

confirm these impressions and examine the possible interaction between heterospecific cue location, group size, and sex.

### Model selection

Model 2, a null model (see Table 6), which incorporated rounds of testing as random effects, did not exhibit improved data fit in comparison to Model 1 ($X_2(2) = 3.15$, $p = 0.207$). Therefore, Model 1 was selected as the most parsimonious null model and no random effects were included into our fixed effects models. Within the fixed effects models, Model 3 was identified as the most parsimonious (LR tests, $p > 0.05$). Furthermore, Model 3 exhibited superior data fit compared to Model 2 (a null model) ($X_2(2) = 148.44$, $p < 0.001$), establishing it as the favored model. Incorporating the location of the heterospecific cue filter paper as a predictor improved our multinomial model predictions and reinforced our observation that the heterospecific cue location affected shelter selection in *B. dubia* cockroaches. Nevertheless, shelter choices were not predicted by sex, group size, or their interaction.

### Selected model

Table S5 reports the relative risk ratios of Model 3 for cue location. When we placed the heterospecific cues filter paper under the lighter shelter as opposed to under the reference darker shelter, the risk of finding individuals under the lighter shelter increased by 14.32 (95% CI [3.35–61.31]) and the risk of finding individuals under any shelter increased by 1.41 (95% CI [0.81–2.45]). Although these risk ratios suggest an effect of the heterospecific cue location on *B. dubia* cockroaches' shelter choice, to understand better the impact of the heterospecific cue location on the probability of finding individuals under the darker shelter, we calculated the AMEs for cue location (Fig. 3).

On average, the presence of the heterospecific cues filter paper under the lighter shelter, as opposed to under the reference darker shelter, decreased an individual's probability of being under the darker shelter by 0.112 (95% CI [−0.177 to −0.495], Z = −3.48, $p < 0.001$), but increased its probability of being under the lighter shelter by 0.090 (95% CI [0.052–0.128], Z = 4.64, $p < 0.001$). However, having the filter paper under the lighter shelter did not influence an individual's probability of being out (0.023, 95% CI [−0.031 to

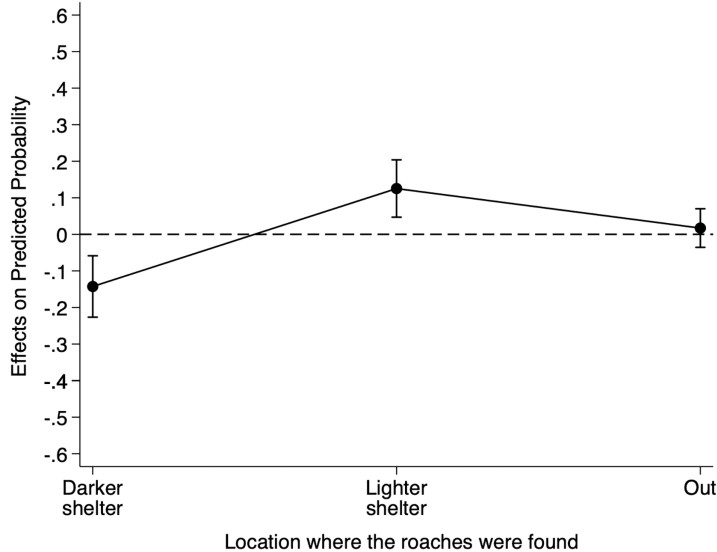

**Figure 3 Discrete change of heterospecific cue location on the probability that *dubia* roaches were in the darker shelter, lighter shelter, or not in any of them (Out) in Experiment 3.** Discrete change (average marginal effects) of heterospecific cue location (from the reference darker shelter to the lighter shelter) on the probability that *dubia* roaches were in the darker shelter, lighter shelter, or not in any of them (Out) in Experiment 3. The dots represent the average marginal effects, and the error bar represents the 95% confidence intervals. The dashed line represents the absence of an effect.

0.078], Z = 0.84, *p* = 0.403). In short, heterospecific chemical cues under the lighter shelter decreased the probability of finding *B. dubia* cockroaches under the darker shelters but significantly increased the probability of finding them under the lighter shelter.

## DISCUSSION

In our first experiment, *Blaptica dubia* cockroaches demonstrated a strong preference for dark shelters with conspecific cues over darkness-matched shelters with no chemical cues. However, neither sex nor trial had any effect on the probability of finding an individual under a shelter with a conspecific cue. In our second experiment, when giving *B. dubia* cockroaches the choice between a darker and a lighter shelter, the location of the conspecific cue filter paper substantially impacted individuals' behavior. When the conspecific cues were placed under the lighter shelter (in comparison to the reference of the cues being under the darker shelter), the probability of finding an individual under the lighter shelter increased. Neither group size (one or three individuals), sex, nor their interaction affected the probability of finding an individual under the darker shelter, the lighter shelter, or out.

In our third experiment, the location of the heterospecific cue filter paper was also shown to impact *B. dubia* cockroaches' shelter choice behavior. In comparison to the reference condition of having the heterospecific cues under the darker shelter, when the filter paper containing heterospecific cues was placed under the lighter shelter, the probability of finding an individual under the darker shelter decreased significantly. Conversely, the probability of finding an individual under the lighter shelter increased.

Furthermore, as in the second experiment, we could not detect an effect of group size, sex, or their interaction, on the probability of finding *B. dubia* cockroaches under the darker shelter, the lighter shelter, or in no shelter.

These results revealed that *B. dubia* cockroaches used conspecific cueing in making their shelter choice, a mechanism widely observed in natural habitat choice in a wide range of species. In the absence of direct, individually-acquired information about the surrounding habitat and the availability of important resources like shelter or food within that habitat, it can be adaptive to rely on the public information made available by conspecifics and their behavior (*Valone & Templeton, 2002*; *Danchin et al., 2004*; *Kendal, Coolen & Laland, 2004*; *van Bergen, Coolen & Laland, 2004*; *Bonnie & Earley, 2007*; *Canonge, Deneubourg & Sempo, 2011*). Furthermore, it was clear from the third experiment that individuals' shelter choice behavior was influenced by chemical cues of another species. Although not as widely documented as conspecific cueing, heterospecific cueing can be an important factor in shelter or habitat choice in animals (rodents: *Cassaing et al., 2013*; snakes: *Waye & Gregory, 1993*; birds: *Tolvanen et al., 2020*). Even slight attraction to heterospecific cues can lead to important mixed-species grouping dynamics, as seen in terrestrial isopods, *Porcellio scaber* and *Oniscus asellus* (*Broly et al., 2016*) and in bombardier beetles of genus *Brachinus* (*Schaller et al., 2018*).

Our results from Experiments 2 and 3 suggest a weaker effect of heterospecific chemical cues on *B. dubia* cockroach shelter choice in comparison to conspecific chemical cues (Figs. 2 and 3). However, we were unable to determine whether there is a real difference between conspecific and heterospecific chemical cues used by *B. dubia* cockroaches in their shelter choice. There were numerous potential confounds for such a comparison in our experiments, including the densities of the roach colonies, the degree of exposure of the filter paper to chemical cues, and the different shelter darkness levels. Future work will aim to control these factors in direct tests of shelters containing *B. dubia vs. B. craniifer* cues.

In our earlier study testing the effects of different group sizes on darker *vs.* lighter shelter choice in *B. dubia* cockroaches, we found that the sex of the individuals had minimal effect on shelter choices (*Freeberg & Fiset, 2023*). We also found little effect of sex on shelter choices in the context of conspecific or heterospecific cues in this study. This lack of an effect of sex is interesting when considering the anatomical and biological differences that exist between female and male *B. dubia*. Females of this species are much heavier than males so it can reasonably be inferred that females are likely to move differently or at different rates than males (though this awaits testing). Sex differences in activity are known for some cockroach species (*e.g.*, *Periplaneta fuliginosa*, *Appel & Rust, 1986*; *P. americana*, *Nicolis et al., 2020*). Female and male *B. dubia* cockroaches do differ in their preferences for the size of individuals with which they associate, with males preferring larger individuals and females preferring smaller individuals (*Fisher, 2023*).

Little is known about *Blaptica dubia* behavior in natural settings. They are known to associate with other invertebrate species. For example, over half of samples collected from colonies of the Hemiptera insect, *Triatoma reubrovaria*, contained individual *B. dubia* cockroaches, suggesting a trophic relationship between the two species (*Salvatella et al., 1995*; but see *Durán, Siñani & Depickère, 2016*). We do not know if the seemingly weaker

heterospecific cueing effect we detected here is due to a general weaker heterospecific cueing effect, to the specific novel species tested here (*B. craniifer*), to differences in the relative light levels of the shelters in Experiments 2 *vs* 3, or to some other factor(s). Future studies to address these questions would benefit from testing cues of species with which *B. dubia* naturally interact. This type of relationship between naturally interacting species was demonstrated in recent work with *Brachinus* beetles, in which researchers found that most naturally occurring aggregations contained multiple species of the genus. This study also revealed in laboratory experiments that some species, such as *B. hirsutus*, preferred to aggregate with members of another species (*Schaller et al., 2018*).

Like many social species, cockroach behavior is sensitive to variation in the social environment. For example, paired female German cockroaches, *Blattella germanica*, eat more per individual than do isolated individuals and, even when controlling for total food intake, paired females produce larger oocytes than isolated females (*Holbrook et al., 2000*). Contact with another individual's antennae and/or whole body appears to be important to this "pairing" effect on oocyte development relative to isolated individuals (*Uzsák & Schal, 2013*). Additionally, individual cockroach running speeds in different layouts of mazes have been shown to vary depending on whether individuals run by themselves, with a conspecific, or in a context where an "audience" of conspecifics is visible but not themselves running in the maze (*Zajonc, Heingart & Herman, 1969*; *Halfmann, Bredehoft & Hausser, 2020*).

Here we have documented *B. dubia* cockroaches' sensitivity to chemical cues of conspecifics (corroborating findings in a wide range of cockroach species) and also of heterospecifics. Chemical cues of other roaches are an important part of the variation in their social environmental context and can powerfully shape aggregations of individuals and drive processes like species segregation in space (*Leoncini & Rivault, 2005*). Moving forward, we need to assess the relative roles of chemical cues compared to other cues (such as visual or tactile cues) in shelter choice behavior. Furthermore, studies show that individual shelter choices and group shelter choices do not always correspond with one another but depend heavily upon physical and social environmental contexts (*Laurent Salazar et al., 2017*; *Nicolis et al., 2020*). Thus, repeated testing of the same individuals in diverse environmental contexts—including variation in shelter darkness and variation in concentrations of chemical cues—could prove essential to understanding individual decision-making in shelter choice (*Planas-Sitjà & Deneubourg, 2018*). Additionally, we would like to obtain further data on *B. dubia* cockroach behavior related to searching for shelter. Namely, are individuals choosing shelters based on the random encountering of chemical cues within the test arena or as a consequence of active assessment and exploration of their environment?

## CONCLUSIONS

In a series of three experiments, we found that *B. dubia* cockroaches' shelter choices were heavily influenced by whether those shelters contained conspecific (Experiments 1 and 2) or heterospecific (Experiment 3) chemical cues. Individuals were more likely to choose a shelter that contained conspecific cues compared to a darkness-matched shelter that did

not contain those cues and were more likely to choose less-preferred lighter shelters when chemical cues were placed under those shelters. These studies reveal *B. dubia* cockroaches' sensitivity to stimulus variation in their social environment. Indeed, the effects of the social environment, or of social cues, on individual behavior seen in this study and the others described above all involved genetically unrelated individuals that had social experience with one another over extended time (see also *Jeanson & Deneubourg, 2006*; *Lihoreau, Costa & Rivault, 2012*). Work is needed in a wider range of taxa to understand the interactions among individuals within these groups, and how the relationships that may consequently develop, can structure the organization of social groups (*Hinde, 1976*; *Atton et al., 2014*; *Bergman & Beehner, 2015*). These interactions and relationships include affiliative behavior as well as competition and dominance-related interactions (*Lee et al., 2016*; *Coppinger et al., 2023*), and may be the key to understanding how social contexts shape decision-making and behavior in social species (*Freeberg & Coppinger, 2023*).

## ACKNOWLEDGEMENTS

We thank Colton Adams, Scott Benson, Heather Brooks, Brittany Coppinger, Eric Frazier, Hwayoung Jung, Jeff Kelly, Steven Kyle, Harry Pepper, and particularly Mariano Calvo Martin and two anonymous reviewers, for helpful comments on earlier drafts of this manuscript. We thank Daniel Blaisdell, Bradley Madrid, and David Vo for help with data collection.

### Funding

This work was supported by funding from the Department of Psychology, University of Tennessee—Knoxville. The funders had no role in study design, data collection and analysis, decision to publish, or preparation of the manuscript.

### Grant Disclosures

The following grant information was disclosed by the authors:
Department of Psychology, University of Tennessee—Knoxville.

### Competing Interests

The authors declare that they have no competing interests.

### Author Contributions

- Todd M. Freeberg conceived and designed the experiments, performed the experiments, analyzed the data, prepared figures and/or tables, authored or reviewed drafts of the article, and approved the final draft.
- S. Ryan Risner performed the experiments, authored or reviewed drafts of the article, and approved the final draft.
- Sarah Y. Lang performed the experiments, authored or reviewed drafts of the article, and approved the final draft.

● Sylvain Fiset analyzed the data, prepared figures and/or tables, authored or reviewed drafts of the article, and approved the final draft.

## Data Availability
The data, Stata script, and analyses are available at Open Science Framework: Fiset, Sylvain, and Todd M Freeberg. 2023. "Conspecific and Heterospecific Cueing in Shelter Choices of Blaptica Dubia Cockroaches." OSF. December 11. DOI 10.17605/OSF.IO/3Y79N.

## Supplemental Information
Supplemental information for this article can be found online at http://dx.doi.org/10.7717/peerj.16891#supplemental-information.

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
