# Peer review of "Conspecific and heterospecific cueing in shelter choices of Blaptica dubia cockroaches"

_PeerJ, doi:10.7717/peerj.16891_

## Round 0.1 · original submission · Major Revisions

I have received three quite disparate reviews of your work. One reviewer found the paper very unclear and recommended rejection. It should be noted that even this reviewer indicated an appreciation for your research question. Another reviewer had a very positive assessment of your paper but recommended some helpful suggestions and the third was somewhat in between, suggesting the need to tone down the novelty claims and incorporate more existing literature (a point also shared by the others). This reviewer also made numerous comments regarding areas of the design that need explanation or clarification. My own read is somewhat more consistent with the most positive reviewer. I found the introduction to be quite clear and well written. I disagree somewhat that the experimental design and analysis were overly complicated. I found the design quite clear and sensible. However, I am confused by the “sets” referred to on line 221. Wouldn’t you have individual roaches nested within their groups (n = 24)? It is important that you attend to the many helpful comments and suggestions of the reviewers who have more expertise in this area than I do.

·

Basic reporting

The premise of this research is extremely interesting, what a the roles of individual information and public information (be it homo- or hetero- specific) during shelter selection of a (social?) arthropod.
However, I had many problems:
The manuscript was extremely difficult to understand, the construction of the phrases were confusing, i Think the manuscript would benefit from an English editing.

What was said in the text didn't always correspond to what i saw on the figures and tables. Table 2 was difficult to understand.
Although I'm not questioning the validity of the results, I had a very hard time understanding them.

Experimental design

Although it is an interesting original research, the questions are not well defined. And there are some experimental gaps not answered (e.g. what are the nature of the cues) in neither in the Introduction nor the Material & Methods.

Material & Methods section was really difficult to understand, at the end of this section and even after the Results section i steel didn't understood completely the experimental procedure.

Validity of the findings

I was very difficult to judge since the Result section was not compressive, the statistical analysis were not fully interpretative. And i belie that other, more straight forward, tests would had been easier to convey the information and the validity of the observed outcomes.

Finally, note that from what i gathered from the manuscript, i believe that the findings presented in this research are of scientific significance. However, i found that the authors lacked to express how and why these findings are relevants to the scientific community.

Reviewer 2 ·

Basic reporting

In this paper authors study whether cockroach Blaptica dubia use social information (conspecific vs heterospecific chemical cues) during shelter selection, and whether the use of it depends on sex or quality of the shelter. While the scientific question is sounding, the study is not new as authors claim in the introduction/discussion. This study kind of reproduces the work of Rivault and Cloarec with another species of cockroach. While I totally support reproducibility in science, the novelty claims in this manuscript should be revised (see details below). I give a non-exhaustive list of papers on this topic (just a few on the top of my mind). Regarding methods, I acknowledge that authors did a very good effort with the experiments, with a very high sample size and controlling for several factors (sex, cues, etc). However, I don’t understand why they chose such a complicated experimental design. When reading the manuscript I had the feeling that authors needed over-complicated GLM structures and analyses to make up for the fact that they did not use a proper experimental design. For instance, the use different shelters, grouping and test design in each condition, and the experimental timing, in which adaptation time (increasing light period) accounts for 90% of total experimental trial time, make very difficult to evaluate the claims regarding the comparison between conditions. While each condition could potentially stand on its own, their experimental design does not allow a non-biased comparison between these conditions.

In the Introduction, my main issue is the novelty claims of the study. Authors seem to start acknowledging that the use of social cues (aggregation pheromone or hydrocarbons as referred in literature), and light avoidance (which is quite obvious given that their activity period is nighttime) in cockroaches has been well established in the literature. However, later on seems that the fact that B. dubia can use hydrocarbons for aggregation is not well known or unexpected, and that the preference of B. dubia for dark places was recently discovered in 2023. In my opinion, the present study corroborates that B. dubia can use social cues as it was expected, and brings support to past literature on this topic. Please see a short list of papers that may be of interest:
Rivault & Cloarec 1998 – Animal Behaviour (Cockroach aggregation: discrimination between strain odours in Blattella germanica)
Amé et al 2004 – Animal Behaviour (Cockroach aggregation based on strain odour recognition)
Rivault et al 1998 – J. of insect physiology (Cuticular extracts inducing aggregation in the German cockroach, Blattella germanica (L.).)
Saïd et al 2005 – J. of insect physiology (Cuticular hydrocarbon profiles and aggregation in four Periplaneta species (Insecta: Dictyoptera).)
Lihoreau et al 2012 – Insectes Sociaux (The social biology of domiciliary cockroaches: Colony structure, kin recognition and collective decisions)
Leoncini and Rivault 2005 – Ethology (Could species segregation be a consequence of aggregation processes? Example of Periplaneta americana (L.) and P. fuliginosa (Serville))
Canonge, et al 2009 – J insect phys (Self-amplification as a source of interindividual variability: shelter selection in cockroaches)
Laurent Salazar, et al. 2017 – Roy Soc Open Science (Group choices seemingly at odds with individual preferences)
Planas-Sitjà, I. and Deneubourg, J.-L. (2018) – biology open (The role of personality variation, plasticity and social facilitation in cockroach aggregation)
Nicolis, et al. 2020 – insectes sociaux (Sexual group composition and shelter geometry affect collective decision-making: the case of Periplaneta americana)

Experimental design

In general, Methods could benefit from some re-writing. It is quite confusing sometimes whether authors were testing groups or individuals. Also, there are many imprecisions in the text, such as “at least three days”, “yellow or orange”, “blue or purple”. These comments need to be precise, as are important for interpretation of the results, (e.g., colour is important as red light is perceived differently than blue light; or days between experiments or how long a filter paper is left in a box affects the experiment).

I have some specific concerns about the methods.
Grouping and tests: I don’t understand why group cockroaches by 4, if they are tested separately later, or 1 individual + group of 3. If authors did not care about repeatability of behaviour, then why test same individual 6 times? This adds complexity to their GLM analysis with nested individuals within groups etc, that was not needed.

Gradual intensity of light: I understand that authors gave some time to explore the arena in darkness condition. However, I was surprised to see that the adaptation period is 30 minutes, and the full-light phase only 3 minutes. This design means that cockroaches did their choice probably under dark or semi-dark condition (when light was still turning on). In that case, the perceived light inside shelters is different from the one measured with full light. Is difficult to explain the effects of such treatment, and we could potentially argue that what authors observe is an attraction to the ‘soaked’ filter papers, no matter the shelter treatment.

Shelters: Authors are comparing results between treatments but… the colour of the shelters are not consistent between treatments. A blue shelter is considered dark in one treatment, but light in the other.

Validity of the findings

Authors clearly show that B. dubia cockroaches are able to perceive conspecific & heterospecific cues, and are attracted to them. However, because of the concerns highlighted above, it is difficult to assess the validity of the other findings.

Minor comments:
Please be consistent with the use of the term “cockroaches” or “roaches”. I suggest using ‘cockroach’ instead of ‘roach’ for a scientific paper.

In results section, I am not sure that “nested” or “not nested” is the correct term when discussing the significance of a LR test. By definition, LR only allows the comparison of nested models.

Table 3 and 4. Did authors test the model with fixed effects Cue X Sex + group, or Cue X Sex or Cue + Group + Sex? A recent paper shows how sex can play a role in aggregation dynamics of cockroaches (Nicolis et al 2020).

L95: I suggest taking a look at Nicolis et al 2020 too.
L98-106: please rephrase, I did not understand whether authors were testing groups, individual…
L143-150: Needs some re-organisation. L147-150 should go before previous sentence to explain what is a ‘retirement’ box.
L152 – 160: Why inter-observer reliability? How many observer did they had? I though it was just one trained research assistant.
L177: Was the foam sheet smaller than the shelter?
In L367 – 374, I think the values reported are not correct (different from table S1).
Legend of figure 1 should be revised (eg, ‘vertical black line’ is incorrect).
L651 – Sensitivity of cockroaches to con-hetero-specifics was already shown; authors show sensitivity in B. dubia, or corroborate previous findings.
L666 – Is not surprising at all, these individuals are gregarious, they react to plenty of smells and pheromones (sexual pheromone, alarm pheromone, etc). Authors in introduction already mention the use of social cues in many non-eusocial animals, why should this be unexpected now?

Reviewer 3 ·

Basic reporting

The authors demonstrate the role of chemical marking in shelter selection in B. dubia through several simple and elegant experiments (3 series). Of particular interest are the experiments (and their results) on the synergy or conflict between marking and darkness of a shelter. The results of the third series of experiments on heterospecific interactions are also very interesting and promising. I would add that interspecific communication is attracting increasing interest.
The introduction presents the framework and objectives of the work in a clear way, that is accessible for a broad readership. The different hypotheses are clearly presented in the introduction and the experiments (1,2,3) and analyses to validate them or not. The experiments are well designed and are carefully conducted, while the data are analyzed thoroughly and the results support the conclusions. I really appreciate this work and its very interesting results. Moreover, I find the problematic of this paper of sufficient general interest to justify a publication in Peerj.
I recommend publication of the article, and I apologize for the delay in reviewing the manuscript.

I have the following minor questions/comments/suggestions that I would like the authors to answer:

Keywords
Blaberus cranifer, aggregation (?), chemical marking should be added

Literature references
My main comment: some references to chemical marking in cockroaches could be added: e.g.
Rivault, C, Cloarec & Sreng L (1998) Cuticular extracts inducing aggregation in the German cockroach, Blattella germanica (L.). Journal of insect physiology, 44, 909-918

Saïd, I., Costagliola, G., Leoncini, I. & Rivault, C. (2005) Cuticular hydrocarbon profiles and aggregation in four Periplaneta species (insecta: Dictyoptera). Journal of Insect Physiology 51, 995–1003

Hamilton, J.A., Wada-Katsumata, A. & Schal C. (2019) Role of Cuticular Hydrocarbons in German Cockroach (Blattodea: Ectobiidae) Aggregation Behavior
Environmental Entomology, 48, 546–553.

And concerning heterospecific aggregation: e.g.
Goodale, E., Beauchamp, G. & Ruxton, G. (2017). Mixed-Species Groups Of Animals: Behavior, Community Structure, And Conservation. Cambridge, MA: Academic Press.
(book devoted mainly to birds and mammals)

Boulay, J., Aubernon, C., Ruxton, G.D., Hédouin, V., Deneubourg, J.-L & Charabidzé, D. (2019) Mixed-species aggregations in arthropods. Insect Sci. 26, 2–19 (review)

Schaller JC, Davidowitz G, Papaj DR, Smith RL, Carrière Y, Moore W (2018) Molecular phylogeny, ecology and multispecies aggregation behaviour of bombardier beetles in Arizona. PLoS ONE 13(10): e0205192.
Methods
They are described with sufficient details.
Suggestion/minor comments:
-some procedures/materials are the same for all 3 experiments. I would avoid repetition and group as much information as possible under "General Methods" (e.g. illuminometer, the two shelters were considerable darker than open area,....).
-The dimensions of the set-up and shelters for series 2 and 3 are the same as for series 1 (lines 173-177)?
In addition, I'll add a table summarizing all the experiments, with the number of replicates, the lux of each shelter,...
-Concerning the chemical marking :
--Density/number (no. of cockroaches/container or per square centimeter) are around ??? (lines 143-147)
--Are the densities/number of B. cranifeer and B. dubia more or less equal?
L 152 -160: I don't understand: how many observers are involved?
-Marked filter paper: are the faeces removed? (e.g. with a brush)

Analysis & Results
-Figure 1 : the vertical black line represents the predicted mean ?
-Access to data/supplementary: A glossary of abbreviations used in the files is required

Exp 1
A quick analysis of the distribution of the number of times an individual is present in the marked shelter (over the 6 tests and from your raw data) leads me to believe that the variability between individuals is greater than that expected with identical individuals. (e.g. females global mean: 3.6, variance: 2; theoretical variance 1.45 with identical individuals).I therefore think that a "personality or idiosyncrasy" is present.

Exp 2
Bearing in mind what you say about the work of Freeberg and Fiset (2023) (unfortunately, I did not get access to the paper) and Lines 472-474:

The presence of conspecifics (N=3 vs N=1) has a positive effect (positive feedback) in the "Lighter shelter with cue" case. Interestingly (if I've understood the data in the supplementary), the mean probabilities of being in the darker are more or less the same for isolated or grouped individuals, but the distribution for N=3 shows a bimodal (or uniform) distribution and is statistically different from a multinomial/binomial (see female figure PDF).
Figure Number of replicates as a function of the number of sheltered individuals (in darker shelter); blue experimental; orange theoretical.

The difference in populations outside shelters between the "darker with cue" and "lighter with cue" cases could be explained by the average residence times in each shelter?
Exp 3
I did not get access to darkXheterocueing4.dta

Discussion
I find the discussion stimulating and honest by pointing the limitations of this work.
Line 647-651 Among future projects: From my point of view, the influence of the concentration of marking (e.g. length of time of marking) on the cockroach behaviour is also a priority?
Line 633 concerning the relationship between Triatoma and cockroach please check: Pamela Durán, Edda Siñani, Stéphanie Depickère (2016) On triatomines, cockroaches and haemolymphagy under laboratory conditions: new discoveries. Mem Inst Oswaldo Cruz, Rio de Janeiro, Vol. 111(10): 605-613

Experimental design

No comment

Validity of the findings

No comment

Annotated reviews are not available for download in order to protect the identity of reviewers who chose to remain anonymous.

---

## Round 0.2 · Minor Revisions

As you can see, two of the reviewers continue to find the analysis and results sections of your paper quite confusing and suggest alternative approaches to explaining your approach. Please attend to their concerns and additional requests for clarification.

·

Basic reporting

No comment

Experimental design

The authors significantly improved their manuscript. It is easier to read and understand the manuscript. However, I still have major issues, mainly with the experimental procedure, it is not easy to understand ( for example how long a trial lasted?). Moreover, the statistical analysis section of each experiment is a bit confusing as well.

Validity of the findings

No comments

Additional comments

Somme other issues are listed below.
Line 28 and other lines: why do you use dubia and not B. dubia?
Lines 166-167: Why? For those, not familiar with social arthropods, this phrase might confuse.
Line 202: I think that “half of the dubia roaches in the trials”, is easier to understand.
Lines 293-294: The 50/50 sex-ratio of the groups is it similar to those observe in nature?
Lines 341-344: what was this concern?
Table 1: some questions:
Why the light levels (lux) for a same type of shelter are so different between the experiments?
Almost 200 lux of difference within the lighter shelter in experiment 2, is not too much?
In the filter paper columns, why the range is so large? , don’t you think that a three-day coding is less attractive/retentive than a 13 day coding?
Lines 344-354: It is no easy to understand.
Roaches were able to explore the arena and the shelter before the light where turn on?
By “coding the shelter”, you mean putting the hydrocarbon marked filter papers for the “retirement colony container”?
What “After shelter choice coding“ means, the end of a trial? If yes, how long was it?
The experimenter disturbs the roaches while they are in the test arenas or the shelter?
“Odourless baby wipes” for the roaches also? Are sure the no hydrocarbons are left? Why not use soap-water and alcohol?
Lines 482-497: why do you use different light colours? As different light wavelengths might have different behavioural responses, why not to use same colour but with more or less lux?
Line 504-505: But with the cockroaches in it?
Lines 548-552: Why you didn’t use the same plastic sheet as in experiment 2
Line 558: How long a trial lasted?
Lines 637-640: which one/s? All four of them or the last one?
Sometimes you use “non-shelter open spaces” other outside, please chose one.
How do you account for any behaviour induce due to an ovarian cycle?
Results
Regarding all three experiments, why do you only talk about the individuals, and not the groups as a unity? You say X % of individuals ended in one or the other shelter or outside, but how many groups ended with all individuals in the same shelter (a consensus). This missing information could, among other things, infer on the hierarchy of the communication (signals and/or cueing) pathways (chemical, mechanical/thigmotactic, …).
Experiment 1:
How do you explain the fact that the number cockroaches outside the shelter is higher than in the control shelter, and this tends to increase with the number of trials?
Figure 1: Horizontal axis, shouldn’t be round instead of trying?
Lines 687-688: Model 4 fits the data better than the Model 5 (X2(2) = 0.98, p = 0.612) the p value shouldn’t be bellow 0.05? If not, not only I didn’t understand the test (but that’s my problem) you should verify line 637.
Experiment 2:
Lines 927-932: Not quite sure that I understand completely. So globally, the “winner” shelters are: Dark_with_cue > Light_with_cue > Dark_NO_cue > Light_NO_cue? A table other than T3 might help.
Lines 1071-1082: Quite interesting! Why do you think that is, biologically speaking?
Lines 1380-1382: these phrases should be toned down. OK, for a group size of three individuals there is no influence of the group size nor sex, but is this also true for higher population sizes. Furthermore, is three individual representative of a natural/wild population of these cockroaches. And what would be the case if a female is emitting a sex pheromone?

Reviewer 2 ·

Basic reporting

Authors have assessed most of my questions, and I found the manuscript much easier to read and more interesting than before. I believe that the discussion is well adapted to the results found, the introduction acknowledges previous research on the topic, and the methods section presents the experimental design in a clearer way. However, I still have some concerns regarding the results section. The issue for me, as a reader, has nothing to do with the use of statistics or understanding them, but I feel that the message that authors are trying to deliver is not clear. I give here below a detailed explanation. I believe authors could modify (and reduce) the results section to improve clarity quite easily.

Results:
I think authors went a little over-board adding explanations in results sections. After taking a look to other referee comments, I believe there was a lack of explanation about methods used, and also some disagreements with how results were reported compared to results shown in figures and tables. Authors have heavily modified this section, but I am afraid that these modifications failed to convey the results in a clear way. Basically, the rationale of how authors chose the most parsimonious model is quite complicated and convoluted, and it is quite hard to follow, specially with too many details and comparisons. I will try to give an example to explain my point of view.

L382-414: This is how I see the rationale in the text: “In terms of random effects, we tested M1, M2, M3. […] Model 2 is the preferable one, but has more parameters, but has lower AIC, although LR tests confirm model 2 is better than 1, but 3 is not better than 2. In terms of fixed effects, M4 is the most parsimonious, confirmed by LR, with M4 better than M5, and M5 better than M6. However, M3 (?) shows no effect of sex, and M3 was not different from M2, as such M2 is the best, and therefore shelter choice does not depend on sex, which can be seen in fig 1 using M6.”

As authors can see, the rationale is not very straight forward. Authors first compare null models to decide random structure with several methods; then discuss all the differences between models with different fixed effects (neglecting the null model) to point out which one is the most parsimonious, and finally compare all of them with the null model to select the most parsimonious model again. It is quite confusing to read that the “best” model is the one including X and Y predictors, but then stating that the null model has a higher likelihood, and thus those predictors (X and Y) have no effect at the end. For instance, it took me a long time to understand what was the message of paragraph 405-414. I think what authors meant (if I understood correctly), is that Model 2 (null) is the most parsimonious model because including the factor sex does not improve the likelihood of the model. If so, why don’t just say that? On a lesser note, the p-value of a predictor is dependent on the structure of the model, so why discuss results of Model 3 that had a different structure compared with Model 4? No need of that.

I believe an easier way to present the same results would be something like “In terms of random effects, the most parsimonious model was X (see table), which included this and that, because lowest AIC/BIC **or LRT comparison**. Thus, this was the chosen structure of random effects. In terms of fixed effects, the most parsimonious model was MX (**including null and full model in the comparison**), including such predictors (see table). This is well illustrated in figure Y, showing the effect of…”

I highly recommend the use of one method (log likelihood, AIC or BIC) to choose models, the method authors believe is most appropriate for the data and aim of the analysis. Using all methods at once is in general bad practice, as it poses the question of what to do when they are in disagreement. How to justify the use of one or the other in each case? For instance, authors sometimes discuss the selection based on AIC, sometimes AIC and BIC, etc. In any case, seems that the final choice is always confirmed by LRT (and LRT is the only one introduced in methods), so probably they should stick to one of these only. This would simplify the explanation a lot.

Experimental design

I have no issues with the experimental design

Validity of the findings

See basic reporting above

Additional comments

Minor comments:

L181: Is this sentence correct? Maybe “… choices of tested dubia roaches”? Or “… the individual shelter choices were coded as …”
L204: separated?
L216: Just a naif comment… is a pity authors could not find a reference measuring HC concentration on filter papers over time. If they could show that after 3 days the HC concentration on a paper is already saturated, then they could justify that leaving it for 3 or 14 days shouldn’t have a big impact on the results.
L224: lamps were?
L227: This sentence raises more questions than provides answers… Now I wonder what concern was that? I suggest removing this sentence and add, in L229: “we used the timer and gradual light intensity to mimic better natural changes in lightening, and help ensure …” (if my understanding is correct and this was the intended meaning)
L250: “four, six” missing comma?
L289: Just to be clear, authors used sometimes dark blue, and sometimes purple sheeting? Is that correct? At least this is my understanding from that sentence.
L431: what is the post-estimation linear combination? This method is not introduced in methods section (I could not find it at least). Also, is it needed?
L510: This should be moved to methods too.

Finally, with all the additional explanations, the introduction and methods are quite long. Even if there is no word limitation in PeerJ (I think), I provide some suggestions that may help improve the flow of the text:

Cutting off some examples from the introduction (e.g., L62-82 or L89) may facilitate the reading.

L117-154: Description of experiments is very long, and repeats a lot of information given in methods. This section could be reduced, giving only the important information to understand the predictions or hypothesis reported. For instance L121 not needed that groups are tested 6 times, or 2 adults and 2 females (L118), etc… The expectation is “choose conspecific cues over no cues”. A simple sentence such as “In exp. 1 we tested individual preferences for shelters with either conspecific odour (filter paper with conspecific chemical cues) or with no odour (clean filter paper).” I believe that this would make it easier to remember what is being tested in each experiment, and ease the reader into the more detailed description in methods.

In methods, L309-311 and 318-319 basically convey the same information (predictors included).
L446-447: No need of this sentence, this information is already in table description; just add in L449 “group size or sex (table 3)”.
L395-396: The table already gives this information, it is not needed in the text.

Reviewer 3 ·

Basic reporting

Dear Editor, Dear Prof. Vonk
I have read the authors' responses to my comments and have read the new version. I am fully satisfied with the revised version.
I'm convinced by the authors' arguments in their rebuttal letter (including some problems that are not fully solved, e.g. concerning the idiosyncrasy or the residence time).
In conclusion, I recommend the publication of this revised paper.
However, I still have one comment to make about the data on the “Open Science Framework” .
I cannot find (on osf.io) the variable descriptions and SOM (see rebuttal and the links pages 11 and 12). I don’t claim that these information are missing, but a short “user manual” describing the file’s content would be really helpful.

Experimental design

please see my previous review

Validity of the findings

Please see my previous review

---

## Round 0.3 · Minor Revisions

Per discussion with the journal, we're returning this for a Minor Revision. Please upload your final manuscript files.

·

Basic reporting

Authors have significantly improved the manuscript. In particular, I find that the new version, with the “general statistical approach” it helps a lot to understand the Result section.

Experimental design

Experimental design is now more easy to understand:
However I have some questions issues:

Lines 279-281 and Lines 282-285 say the same thing.
Lines 285-287: If I understand correctly, you put the filter paper with the roaches inside the test arena. If that is the case, what happen if an individual is already under a shelter? You introduce a perturbation that is random (is not the same for all tested individuals).
Lines 287-289: you already say this in lines 180-181. Idem for lines 293-294 and lines 297-298.

Validity of the findings

As for the previous section, it is now easy to understand:
Only one issue:
Lines 433-441: This part is confusing, is there a difference between the “global percentage” and the “percentage of individuals”?

Additional comments

Discussion,
Regarding heterospecific aggregation, Information about the phylogenetic distance between these two roaches might be interesting and could give some insight on why they respond positively to heterospecific marking.

Reviewer 2 ·

Basic reporting

Authors have addressed most of my concerns in this review. I believe the results are much clearer and easier to read than the previous version. My only concern, which was also pointed out by reviewer 1, is the lack of compliance with international rules for reporting scientific names. L 172 and other places: the use of “dubia roaches” is not standard. Please change all of them to at least B. dubia roaches, if not B. dubia cockroaches. I personally prefer cockroach instead of the informal word ‘roach’ (a short for cockroach, mainly used in US) in a scientific article, although I won’t argue against it. However, the removal of the genus name is pushing it too far. It is not a matter of confusion within the manuscript (authors’ answer to reviewer 1), is a matter of good practices and following the international rules for binomial nomenclature. If authors address this concern, I recommend this article for publication at PeerJ.

Minor comments:
Personally, I think the explanation of what is an LR test (L216-221), or what is a backward step wise method (L227-229) are not needed, although I won’t object to that, as it just brings more transparency regarding the methods used, and can be useful for young students to better understand such methods.
L68-71: formatting error: text has some grey highlight
L621: Authors could also cite a more recent paper on sexual differences in a closely related species (P. americana) here: Nicolis, et al (2020) Insectes Sociaux (which is already in their reference list).

Experimental design

no comment

Validity of the findings

no comment

---

## Round 0.4 · accepted · Accept

Thank you for making the slight changes suggested by the reviewers.